# Identifying widespread and recurrent variants of genetic parts to improve annotation of engineered DNA sequences

**Matthew J. McGuffie, Jeffrey E. Barrick** *

Department of Molecular Biosciences, Center for Systems and Synthetic Biology, The University of Texas at Austin, Austin, Texas, United States of America

* jbarrick@cm.utexas.edu

**Data Availability Statement:** Code used for running all computational analyses is available in a GitHub repository (https://github.com/barricklab/widespread-recurrent-part-variants) and has been archived In Zenodo (https://doi.org/10.5281/

## Abstract

Engineered plasmids have been workhorses of recombinant DNA technology for nearly half a century. Plasmids are used to clone DNA sequences encoding new genetic parts and to reprogram cells by combining these parts in new ways. Historically, many genetic parts on plasmids were copied and reused without routinely checking their DNA sequences. With the widespread use of high-throughput DNA sequencing technologies, we now know that plasmids often contain variants of common genetic parts that differ slightly from their canonical sequences. Because the exact provenance of a genetic part on a particular plasmid is usually unknown, it is difficult to determine whether these differences arose due to mutations during plasmid construction and propagation or due to intentional editing by researchers. In either case, it is important to understand how the sequence changes alter the properties of the genetic part. We analyzed the sequences of over 50,000 engineered plasmids using depositor metadata and a metric inspired by the natural language processing field. We detected 217 uncatalogued genetic part variants that were especially widespread or were likely the result of convergent evolution or engineering. Several of these uncatalogued variants are known mutants of plasmid origins of replication or antibiotic resistance genes that are missing from current annotation databases. However, most are uncharacterized, and 3/5 of the plasmids we analyzed contained at least one of the uncatalogued variants. Our results include a list of genetic parts to prioritize for refining engineered plasmid annotation pipelines, highlight widespread variants of parts that warrant further investigation to see whether they have altered characteristics, and suggest cases where unintentional evolution of plasmid parts may be affecting the reliability and reproducibility of science.

## Introduction

Engineered plasmids are ubiquitous tools in the biological sciences. They are used for a wide variety of tasks, ranging from routine cloning of recombinant DNA and protein overexpression to reprogramming cells with new enzymes, sensors, and genetic circuits [1–3]. Engineering plasmids by assembling DNA from different natural sources began in 1973 with the

zenodo.7850317). Plasmid sequences are available individually from the Addgene website (https://www.addgene.org/browse/) or for bulk download from Addgene upon request.

**Funding:** This work was funded by the National Science Foundation (IOS-2103208 and CBET-1554179), the NSF BEACON Center for the Study of Evolution in Action (DBI-0939454), and the National institutes of Health (R01GM088344). The funders had no role in study design, data collection and analysis, decision to publish, or preparation of the manuscript.

**Competing interests:** M.J.M. is an employee of Plasmidsaurus.

construction of plasmid pSC101 [4]. Chemically synthesizing DNA sequences and introducing them into plasmids has now been commonplace for decades [5]. Many plasmids have been passed from researcher to researcher, and their genetic parts have been copied and remixed, practices facilitated by plasmid repositories [6–8]. The net result is that the genetic components on any plasmid used in a laboratory today often have long, circuitous, and usually incompletely known histories. It has only been standard practice to check the sequences of certain pieces of plasmids, such as by Sanger sequencing a gene of interest inserted by a researcher into a vector backbone, to validate that they are present exactly as designed. Large portions of these plasmids, including origins of replication and antibiotic resistance genes that are critical for plasmid maintenance, are typically assumed to be immutable or to have only sustained mutations with no effect on their performance.

Recently, DNA sequencing has become much more affordable and high-throughput [9, 10]. Computational pipelines have been developed for assembling accurate and complete plasmid sequences [11–13], and researchers now have complete information about pieces of plasmids that were rarely sequenced in the past. These full plasmid sequences reveal that there are often discrepancies, usually of one to a few nucleotides, between the actual parts on a plasmid and their expected, canonical sequences. Plasmid DNA sequences need to be annotated with information about the genetic parts they contain so that their contents can be checked. Annotation programs, such as PlasMapper [14], and commercial software, like SnapGene, tolerate some variation in the matches they report to the consensus sequence for a genetic part in a database. However, they do not alert a user when they encounter these imperfect matches, which may obscure changes in the sequence of a part that have functional consequences. We recently developed a plasmid annotation tool, pLannotate [15], that reports the nucleotide identity of imperfect matches so users can evaluate parts that are not in agreement with the reference sequences.

When a researcher encounters a change from the consensus sequence for a critical genetic part, they are confronted with questions and choices. Should they use the plasmid "as is" or spend time trying to correct the change? Does the change matter for the function of the genetic part? Was the change an edit that was introduced by a prior researcher for some forgotten purpose or was it due to a spontaneous mutation?

Unfortunately, there is no comprehensive central repository of genetic part sequences that a researcher can consult to answer these questions. Databases like iGEM's Registry of Standard Biological Parts [16], the Joint BioEnergy Institute's Inventory of Composable Elements (JBEI ICE) [17], and SynBioHub [18] contain many plasmid and genetic part sequences. However, they are not fully curated and are known to also contain spurious and incorrect information [19]. GenoLIB [20] and the related SnapGene database are computationally and manually compiled databases of a fundamental set of 293 common plasmid parts. They include multiple, curated entries for major families of related parts (e.g., different aminoglycoside resistance genes), but do not attempt to capture the functional implications of more subtle sequence variation. Only specialized databases reach this level of precision (e.g., FPbase for fluorescent proteins) [21]. These resources do not exist for most categories of critical genetic parts.

How do new variants of genetic parts found on engineered plasmids originate? Often these changes are due to researchers finding ways to improve or modify part performance. For example, the $lacI^q$ promoter has a single base change that increases its transcription initiation rate by 10-fold relative to the wild-type $lacI$ promoter found in the E. coli genome [22]. Hundreds of fluorescent proteins have been engineered by introducing changes into natural sequences to alter their spectra, stability, maturation rates, and other properties for imaging applications [21]. CRISPR interference (CRISPRi) uses a catalytically dead Cas9 (dCas9) for the purposes of knocking down gene expression [23]. This variant has two mutations that

inactivate the nuclease domain of Cas9, and these mutations have been engineered independently by different groups in Cas9 proteins encoded by different plasmid lineages [24, 25]. Other changes may have purposes that are more difficult to ascertain, such as when researchers introduce silent changes in protein-coding sequences to add or avoid restriction enzyme cut sites to make parts compatible with certain DNA assembly methods.

Further complicating the picture, genetic part variants can also arise due to evolution. Mutations occur when DNA sequences are copied and assembled into new plasmids *in vitro*. When a single-cell transformant of a plasmid is picked, any mutations it harbors become fixed in all of that plasmid's progeny. There are further opportunities for mutations to arise due to *in vivo* errors in DNA replication and repair as plasmids are propagated in bacterial cells. If the mutated plasmid functions as expected by a researcher, and they don't detect or reject a mutation when validating the plasmid sequence, it will be retained. In some cases, selection will even favor mutated plasmids. Engineered plasmids can impose a significant fitness burden on the host cell if they divert resources needed for cellular replication or produce toxic products [26–29]. In these cases, there is a strong selection pressure favoring cells with plasmids mutated in ways that alleviate this burden by reducing or eliminating the designed function [30–33]. Researchers may also impose other types of selection on part/plasmid function, by picking the most fluorescent or largest colonies after a transformation, for example.

Precisely annotating the presence and properties of common genetic part variants—whether they result from undocumented engineering or unintentional evolution—is key to improving reliability and reproducibility in the biological sciences. However, there are many of these variants, and determining which ones to prioritize for time-consuming manual curation and experimental characterization is a challenge. Here, we develop methods for computationally identifying widespread genetic part variants and variants that recurrently arose from convergent engineering or evolution given a large set of plasmid sequences. We use these approaches to create a list of 217 currently uncatalogued genetic part variants that should be prioritized for further characterization and inclusion in annotation databases.

## Results

### Variants of canonical genetic part sequences are common in engineered plasmids

We used pLannotate [15] to annotate 983,436 genetic parts in 51,384 engineered plasmids in the Addgene repository [6, 7] that have been fully sequenced. We found 171,828 examples of parts that did not match their canonical sequences present in the databases used for annotation. These part variants can be broadly classified into 14 different categories (**Fig 1**). As expected, we observed more variants for more common types of parts and for types of parts that generally have longer sequences. The most common non-canonical plasmid parts are protein-coding sequences, with 73,884 total variants observed, which are comprised of 10,406 distinct variant sequences (**Fig 1A**). The part type that had the next greatest number of variants was origins of replication (46,677 observations of 607 distinct variant sequences), and the third most common variant type was promoters (24,319 observations of 905 distinct variant sequences).

Variants of protein coding sequences and origins of replication are relatively close in sequence to their database counterparts. Variants of smaller parts, such as promoters or protein binding sites, exhibit higher relative levels of sequence divergence (**Fig 1B**). Some of the variants we found are known but not differentiated in current databases used for plasmid annotation. For example, pLannotate and SnapGene currently have a single database entry for the ColE1 plasmid origin of replication, which is the pBR322 variant, the sequence found in a

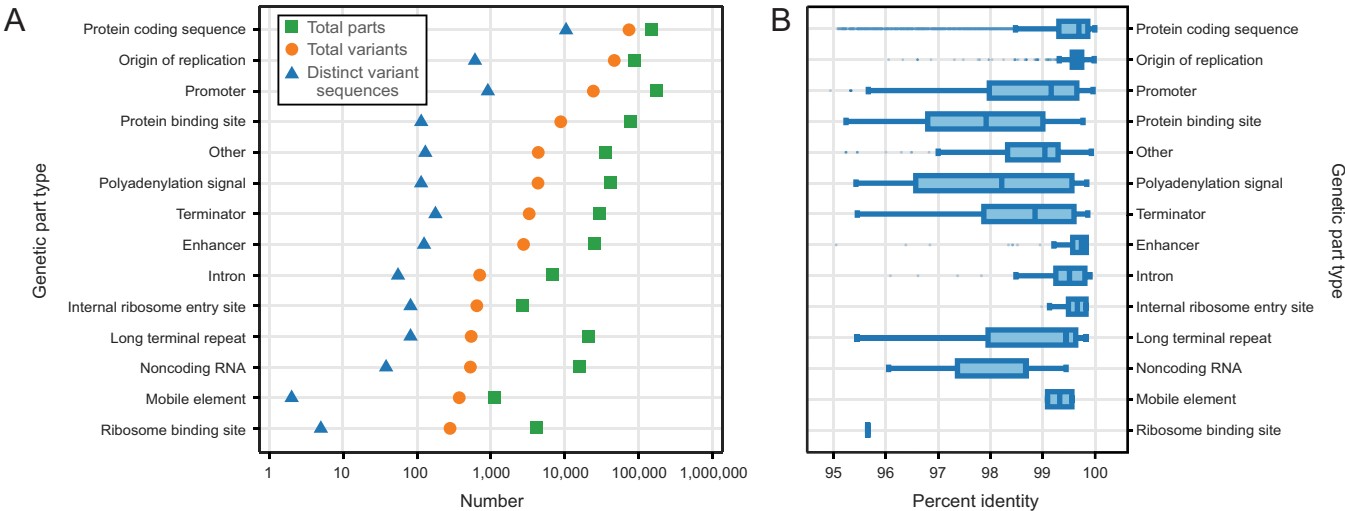

**Fig 1. Many non-canonical genetic parts are found on plasmids.** (A) Overall representation in Addgene plasmids of genetic part variants with sequences that differ slightly from canonical features present in annotation databases. Within each part type, the total number of genetic parts (green squares), total number of genetic parts that are variants (i.e., differ from the canonical sequence) (orange circles), and number of distinct genetic part variant sequences (i.e., counting each unique sequence that differs from the canonical sequence one time) (blue triangles) are plotted. Part types are sorted in descending order by the number of total variants in each category. (B) Distributions of percent identity between distinct genetic part variants in each category and their canonical sequences. Boxes represent lower and upper quartiles (the interquartile range). Vertical lines within each box are medians. The whiskers correspond to 1.5 times the interquartile range. Points are outliers outside this range.

natural plasmid. However, most plasmids contain the engineered pUC19 variant of this origin, which includes a single point mutation that increases plasmid copy number by a factor of about 10-fold [34, 35].

## Some widespread genetic part variants are found on plasmids created by many different labs

The sheer number of plasmid part variants is a challenge for improving plasmid annotation. Our goal is to determine which variants should be catalogued and prioritized as candidates for further investigation, better documentation, and inclusion in annotation databases. The naïve approach would be to catalog all previously undocumented variants, but this is not practical. Engineered plasmids experience severe population bottlenecks when they are constructed and propagated in the laboratory. When plasmids are transformed into a population of cells, typically only a single plasmid enters a successful transformant. It is also standard practice to re-streak cells and isolate a colony derived from a single cell when obtaining a new plasmid from another researcher or from a repository. Therefore, many part variants may be a result of recent genetic drift (fixation of mutations due to chance) caused by these extreme population bottlenecks. Cataloging these "random" variants is not likely to be particularly informative, especially if they are found in just one or a few plasmids.

One might, therefore, propose documenting part variants with the most overall observations. However, this strategy still encounters the same issue. Most variants are found on sets of plasmids deposited by just one or two labs (**Fig 2A**), and some of these variants have become prevalent due to chance (**Fig 2B**). These cases typically occur when a single lab deposits a collection of hundreds of related plasmids that all share the same unique variant of a genetic part. For example, one lab deposited 597 highly similar plasmids, which includes their general lab plasmids as well as a subset used for expressing human SH3 domains [36]. These plasmids all share a single base change in the ColE1 origin of replication. This mutation was almost

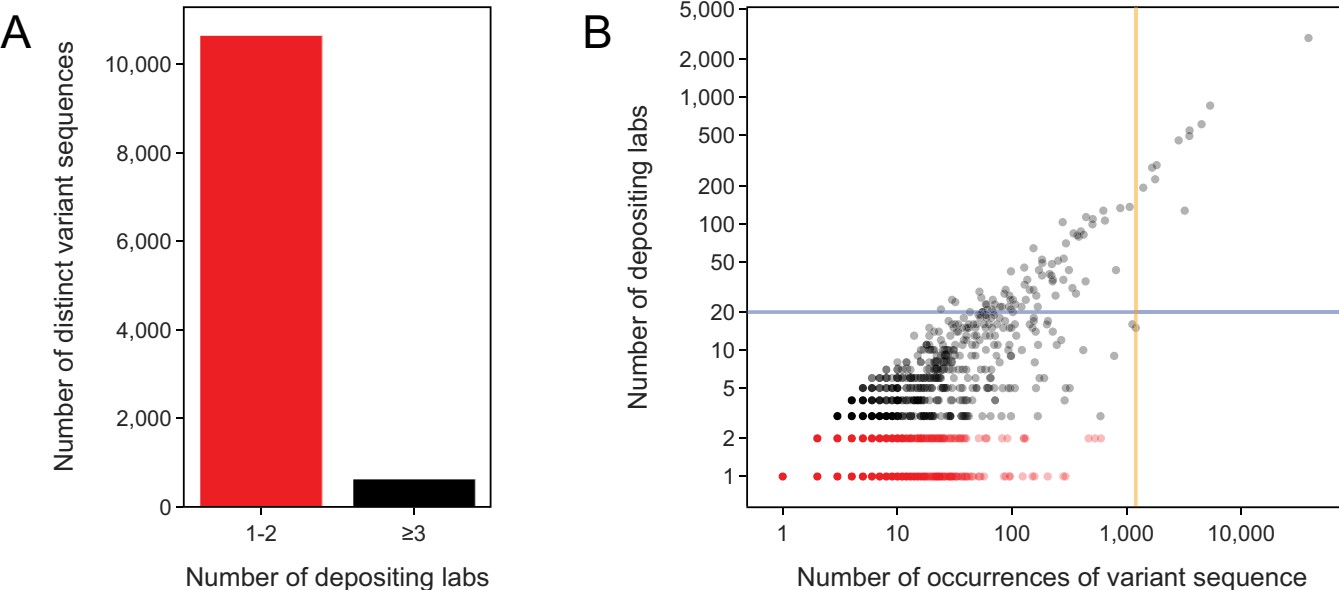

**Fig 2. Most genetic part variants are found in plasmids from one or two labs, but some are more widespread.** (A) Total number of distinct variant sequences found in plasmids from one or two depositing labs (1–2) versus found in plasmids from three or more depositing labs (≥3). (B) All genetic part variants plotted by how many times they were observed versus the number of labs that deposited a plasmid with that variant. The blue horizontal line at 20 labs is the minimum threshold we used for selecting variants that were widespread. The orange vertical line at 1,205 variant observations is the cutoff above which we did not perform the authorship analysis to find cases of convergent evolution or engineering.

certainly present in the backbone of an ancestral plasmid they inherited, and its propagation does not seem to be intentional. Even though this variant is the most common origin of replication variant measured in terms of the gross number of observations (besides the canonical pUC19 variant), we would assign it a relatively low priority for characterization since it appears to be a one-off mutation that was unintentionally cloned into one set of related plasmids.

While deciding which variants to prioritize based on their raw frequencies may not be particularly useful, we believe that cataloging variants found in plasmids deposited by many independent labs does have value. In this case, these variants may also have arisen due to chance in a single progenitor plasmid, but this event likely occurred years or decades in the past, so the potential impact has spread such that it could be affecting many more researchers and experiments. Therefore, we flagged all 75 genetic part variants found in plasmids from least 20 labs (**Fig 2B**, above the blue horizontal line) for inclusion in our set of high-priority variants of interest.

### Recurrent engineering or evolution of unannotated genetic part variants can be predicted using a design similarity score

Variants that are from a few or a middling number of labs are harder to classify. If a variant appears in unrelated plasmids, it could be an engineered variant that is missing from current annotation databases or an evolved variant that arose more than once in unrelated plasmid lineages. Whether designed or evolved, these recurrent mutations are especially likely to affect the function of a part, so it is a high priority to document these cases even if they are in fewer total plasmids. To identify likely examples of convergent engineering and evolution, we analyzed plasmids as authored works. In the natural language processing and information retrieval fields, inverse document frequency (IDF) [37, 38] is a metric employed to predict shared authorship [39–41]. IDF scores the rarity of a word or phrase by counting the observations within a document and compares that to its relative frequency in an entire corpus of

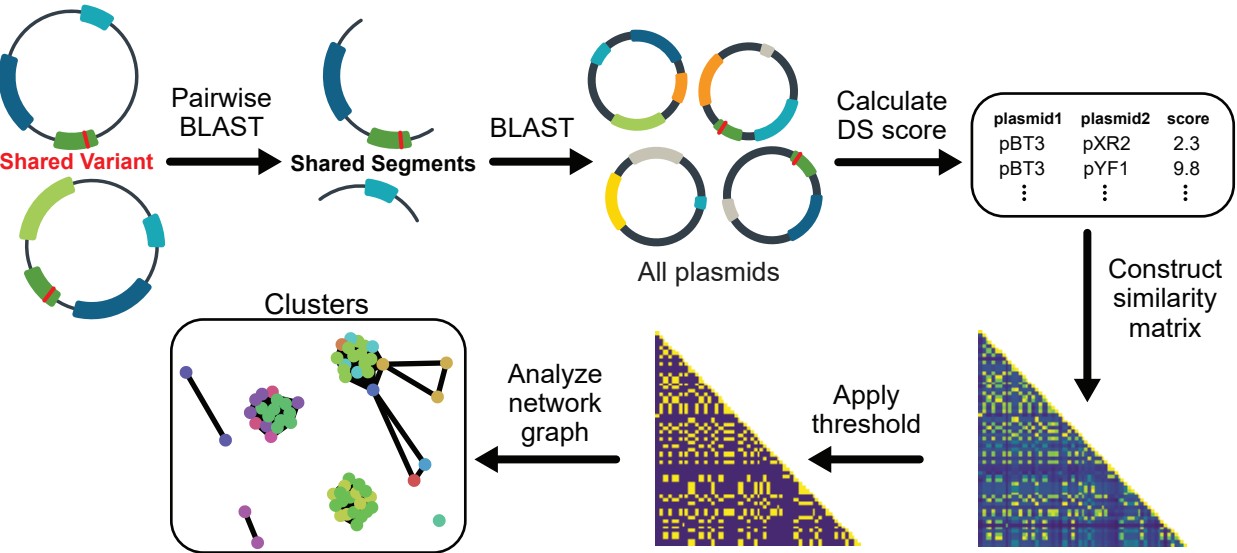

**Fig 3. Method for identifying recurrent genetic part variants that likely arose from convergent evolution or engineering.** All plasmids containing the same genetic part variant are analyzed as a set. Segments shared by each pair of these plasmids are identified and queried against the full plasmid database. The results are used to calculate a design similarity (DS) score between the two plasmids. DS scores for all comparisons are used to construct a network graph of plasmid relatedness. Each separate cluster in the final graph is predicted to represent a set of plasmids in which the variant arose independently.

documents. We created an IDF-inspired metric for use with biological sequences, calculating a quantity that we term the design similarity (DS) score and using it to group plasmids.

Our procedure analyzes sets of plasmids containing the same part variant (shared unique word) for signs of shared authorship (**Fig 3**). We began by identifying all other contiguous sequence segments shared by these plasmids (shared phrases between documents) and tabulating the frequencies of each of these segments in the entire database of all plasmids (how rare the phrases are). We calculated a DS score for each pair of plasmids from these frequencies. Then, we grouped plasmids by constructing a network graph from an adjacency matrix of these DS scores. This step used a score cutoff determined by examining the distribution of DS scores between random plasmids from different labs (**Fig 4**, top). Finally, we divided the resulting network graph into connected clusters that represent groups of plasmids that are unlikely to share the part variant due to common descent or copying of the part.

If multiple distinct authorship clusters are predicted for a variant, it likely had more than one independent origin due to recurrent engineering or evolution. In this case, it should be a priority to document the variant and further characterize whether its function differs from that of the canonical sequence. Because the DS scoring algorithm involves making pairwise comparisons of all plasmids containing a given genetic part variant, it was only computationally feasible for us to apply it to variants with 1205 or fewer observations (**Fig 2B**, left of orange vertical line), which included all variants found on plasmids deposited by fewer than 20 labs that we had not already flagged as being of interest simply because they were widespread. As expected, plasmids sharing a variant that were deposited by the same lab are almost always found within a single cluster at the end of this procedure. This tracks with the intuition that a depositing lab likely recycles their plasmid backbones and pieces of those plasmids for various purposes. In total, 149 of the variants tested using the DS clustering procedure were predicted to occur in two or more author groups. This total includes 7 of the 64 variants tested in this way that were found in plasmids deposited by 20 or more labs.

Using the DS score as a metric has advantages over using a percent identity-cutoff to determine if instances of the same genetic part variant on two plasmids are related (**Fig 4**). Any two

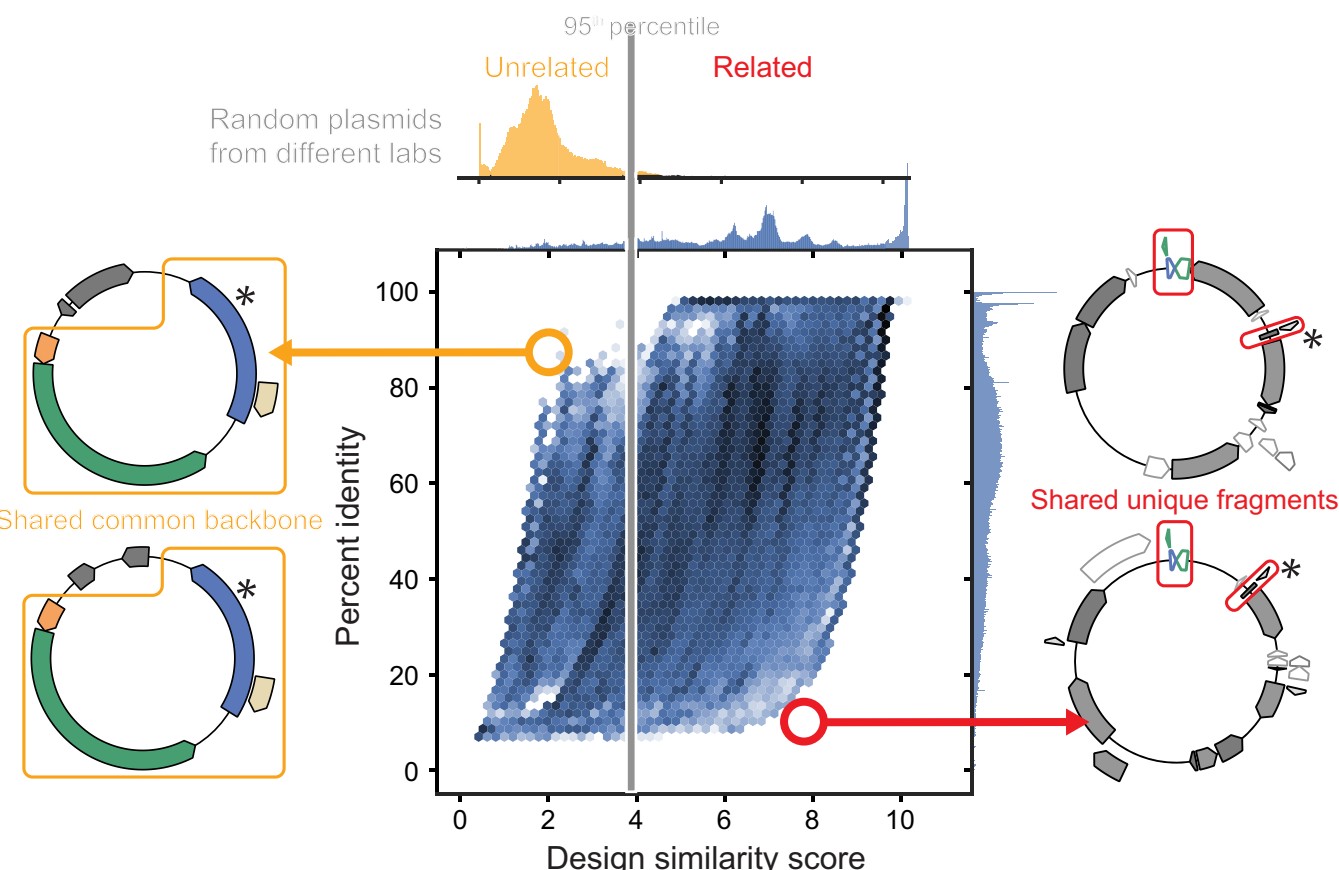

**Fig 4. Design similarity scores reliably identify plasmids that are likely to be related while percent identity does not.** The distributions of DS scores and percent identities for pairwise comparisons of plasmids that share undocumented part variants are plotted. Every plasmid containing a given genetic part variant that was observed 1205 or fewer total times was compared to every other plasmid with that part variant for a total of 7,508,114 comparisons. High pairwise percent identity is not compelling evidence that plasmids are related when they share a commonly used backbone, as illustrated by the plasmid pair shown to the left. The DS score of these two plasmids is low in this instance. Low pairwise percent identity also does not necessarily indicate that plasmids are unrelated, as illustrated by the plasmid pair shown to the right. In this case, a high DS score highlights small, but unique sequences present in both plasmids, which is evidence of shared authorship. Asterisks indicate the location of the shared mutation in the associated genetic part variant that differentiates it from the canonical sequence in the annotation database. The distribution of DS scores between 100,000 randomly selected pairs of plasmids from different labs is shown above the plot. The grey line indicates the 95th percentile of the distribution, which was used as the score cutoff for shared plasmid authorship.

plasmids often share extensive stretches of DNA, but this may not actually indicate anything about how related the plasmids are to each other. For example, the ColE1 origin of replication is used in nearly 95% of the plasmids in our dataset, and 62% of plasmids contain β-lactamase as an antibiotic resistance marker. Since these features are widely used, their co-occurrence is not convincing evidence that a pair of plasmids is related, even if they constitute a majority of the shared sequence identity between them (**Fig 4**, left). The DS metric weights features based on their overall rarity rather than their length or context, so that even a small part or cloning scar can be a strong signal of shared authorship (**Fig 4**, right).

## Final list of widespread and recurrent genetic part variants includes known but uncatalogued mutants

We combined the widespread and recurrent part variants we identified into a final list of 217 currently uncatalogued genetic part variants (**S1 Table**). This list includes diverse genetic parts with a wide range of functions that are used for engineering all kinds of organisms (**Fig 5**). For

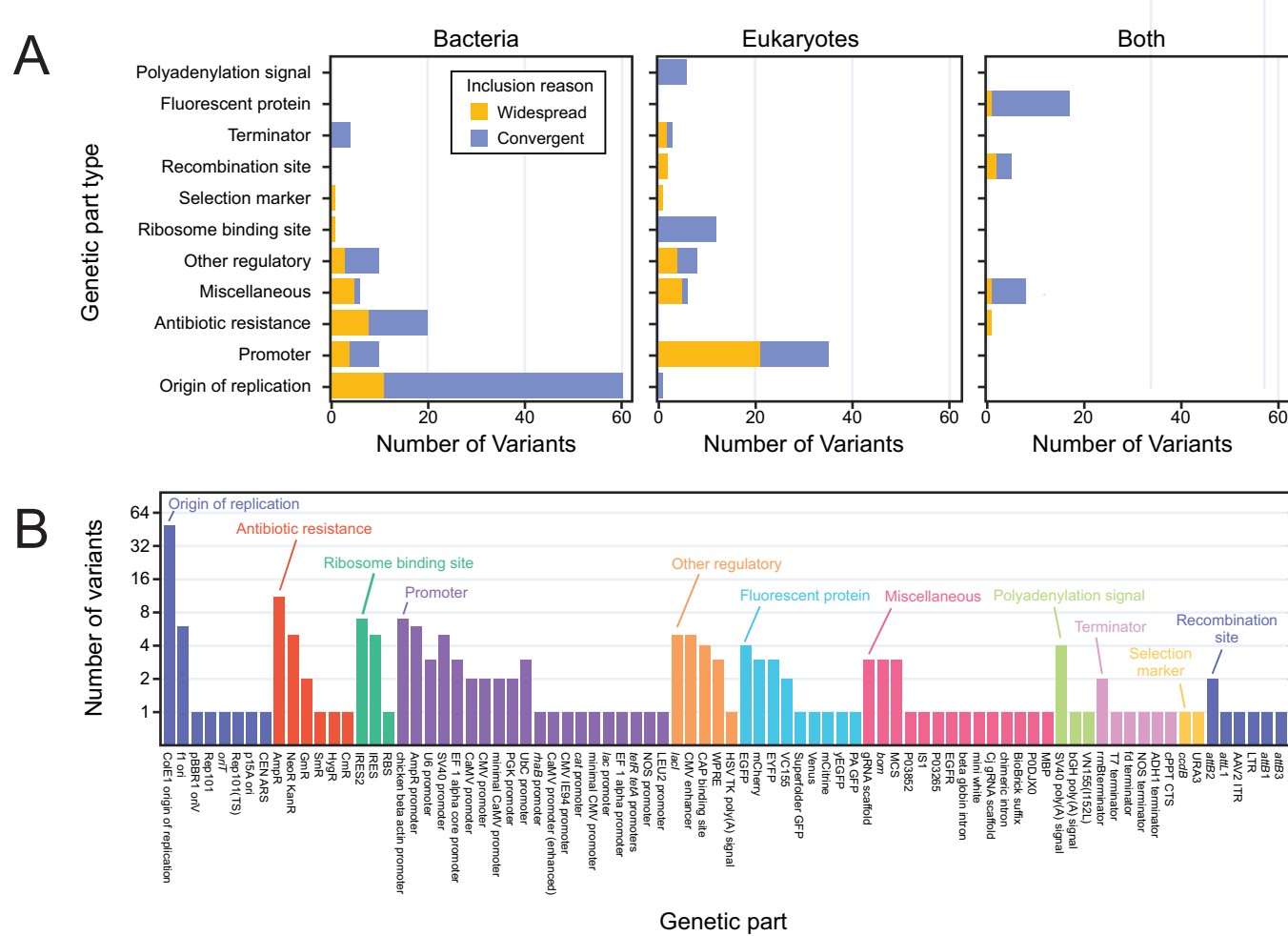

**Fig 5. Uncatalogued genetic part variants to prioritize for characterization and inclusion in annotation databases.** (A) The final 217 variants of interest categorized by part type and by the kind of organism in which the part is typically used. Bars are shaded according to the method by which each variant was judged to be a priority for characterization and annotation: either it occurred in plasmids from ≥20 depositing labs (widespread, orange) or it was in plasmids from fewer labs but there was evidence that it was engineered or evolved multiple times from the authorship analysis (convergent, blue). (B) Names of the canonical parts to which the 217 variants are most closely related. Parts are categorized and sorted by function.

parts designed to function in bacteria, most of the newly identified variants of interest were plasmid origins of replication or antibiotic resistance markers. For eukaryotic parts, promoter variants were most common. Many fluorescent proteins, which function in both types of organisms, were also present in this list of uncatalogued variants not found in current annotation databases.

To validate our inclusion criteria, we looked for cases of known variants that were uncatalogued in the initial annotation databases but were identified by our analysis. The top two variants with 38,693 and 25,995 total observations are the pUC19 variant of the ColE1 origin of replication and TEM-116 β-lactamase antibiotic resistance marker, respectively (Fig 5B). These are both engineered variants that differ from their parent sequences, pBR322 and TEM-1, by one or two bases, respectively [35, 42]. These variants were included in our list because they occurred in ≥20 labs. We also identified one other canonical variant, TEM-171, which was both a frequent and recurrent variant. TEM-171 has one of the two mutations that TEM-116 has relative to TEM-1 [42].

As an example of how these predictions can aid in directing efforts to refine annotations of engineered DNA, one fluorescent protein variant in our list had a clear signal of a recurrent origin due to convergent engineering. Seventeen plasmids with the variant that were deposited by five different labs were from four authorship clusters. This variant is a derivative of enhanced GFP (eGFP) originally described in 1996 by Cormack et al. [43] with additional A164V and G176S amino acid substitutions. This derivative of eGFP is not currently listed in FPbase, and none of the five publications associated with the plasmids containing this derivative mention its provenance or the mutations it harbors [44–48], so their effects on its function are unknown.

## Discussion

It is becoming standard practice for researchers to fully sequence plasmids and other engineered DNA constructs they use in their experiments [11, 49]. These sequences need to be validated by precisely annotating the genetic parts they contain and recognizing unexpected sequence variation in these parts in order to ensure the reliability and reproducibility of science. In the work reported here, we created a list of 217 currently uncatalogued variants of common genetic parts that can be added to databases used by annotation pipelines. These variants are a priority because they are either already widespread in plasmids being exchanged by researchers or they appear to have originated multiple times due to convergent engineering or evolution.

Many of the variants in our final list are in high-copy ColE1-family origins of replication or in antibiotic resistance cassettes that are commonly paired with these origins in *E. coli* vectors used for cloning and replicating DNA. These are by far the most common genetic parts in Addgene plasmids because pUC vectors are used to manufacture high-quality DNA for many applications, ranging from *in vitro* transcription of RNA for biochemical studies to transfection into mammalian cells. Sequence variation in these backbone components might affect cloning success or DNA yields, if a mutation alters plasmid copy number, for example. But, these differences would be unlikely to affect the results of downstream experiments after DNA is isolated from bacterial cells. On the other hand, variants in other origins of replication that we identified, such as the medium-copy p15A origin that is commonly used in plasmids encoding synthetic biology devices meant to function in *E. coli* and the broad-host-range pBBR1 origin that is used for engineering diverse bacteria, are more likely to affect research outcomes. Overall, this logic argues for prioritizing characterization of part variants that are important in the ultimate context in which the DNA will be used, which includes many variants in our final list related to eukaryotic gene expression.

To detect recurrent variants that likely arose multiple times, we developed an approach for grouping plasmids based on signals of shared authorship. Previously, authorship of plasmid sequences has been analyzed from a biosecurity standpoint, with the aim of attributing an unknown plasmid to a specific lab [50, 51]. All of these prior studies analyzed the Addgene plasmid corpus. The first used deep learning to train a convolutional neural network to predict the lab of origin of a plasmid from its DNA sequence [52]. It correctly identified the source lab 48% of the time and the source lab appeared in the top 10 predicted labs 70% of the time. A comparable method, deteRNNt, used recurrent neural networks trained on plasmid sequences and associated phenotype data to identify DNA motifs indicative of different genetic designers [53]. It demonstrated an improvement in accuracy to 70% correct attribution to one lab among 1,300 in the dataset. An alternative approach, PlasmidHawk [54], opted to not use deep learning, citing the higher accuracy and higher interpretability of sequence alignment-based techniques compared to machine learning approaches. Their approach had 76% accuracy in

identifying the lab that deposited an unknown plasmid and could precisely single out the signature sub-sequences responsible for a prediction. Notably, this study used an approach similar to our own where they down-weighted observations of sequence segments that are frequent in the overall dataset, though their metrics differ from our IDF-inspired design similarity score.

We had to infer shared authorship of plasmids to predict when a variant had arisen multiple times because the cloning history of most plasmids is not fully known. Ideally, one would be able to track the provenance of plasmids and their parts using the scientific literature and/or metadata in plasmid repositories to understand which changes to the sequence of a genetic part were intentional and when and how many times they were introduced or arose due to mutations. QUEEN is a recent framework which proposes to record traceable linages of engineered plasmids by having researchers meticulously document their construction process and store this information as metadata in GenBank flat files [55]. Addgene is now encouraging researchers to use QUEEN when submitting new constructs. If this or a similar metadata format for tracking how engineered DNA sequences have been copied, remixed, and modified is widely adopted, it will be very useful for tracking the engineering and evolution of plasmids in the future. Many scientists who performed foundational research creating key plasmid backbones and genetic parts in the early days of recombinant DNA technology are retired or will be soon. It would be extremely valuable if the community could also capture or reconstruct their knowledge of earlier plasmid construction efforts.

pLannotate and other plasmid annotation pipelines use BLAST to find matches to genetic part sequences in a database. This simple approach has some potential shortcomings with respect to variant detection and prediction. One is that BLAST matches may not detect instances of a part or properly delineate their extent when there are mutations at or near its ends. For example, if a bacterial promoter variant has a mismatch in the −35 box at the end of the canonical promoter core sequence and this is also where the part sequence in a database ends, the BLAST hit may only match the downstream part of the promoter. This could result in reporting an incomplete match that is not recognized as a variant or potentially no match at all. Compounding this problem is the issue that some types of genetic parts and important functional variants of these parts can be defined on multiple, overlapping scales. For a bacterial promoter, the database sequence could be just the core element containing the −10 and −35 boxes, or it could be an extended element that includes upstream sequences such as UP-elements [56] or adjacent cis-regulatory elements. Computational matching methods that force extending alignments to the boundaries of part sequences and expert curation of how a core part and elaborated variants of that part are related could help annotation programs deal with these difficult cases.

Ideally, we would be able to provide annotation programs with detailed information to accompany the sequences of the 217 high-priority variants we identified, including their provenance and functional characteristics. It may be possible to trace more of our variants of interest to existing publications in which a researcher engineered mutations on purpose. However, this will require analyzing hundreds or thousands of publications. Since some variants are bound to be the result of *de novo* mutations in the laboratory, these searches will sometimes come up empty. In these cases, one needs to test whether and how the performance of the part variant differs from the canonical sequence and associate that information with the database sequence. Such efforts will take years of expert curation and laboratory experiments by a community of scientists. A framework is needed to centrally collect and organize this information and encourage community participation. FBbase is an outstanding example of continuous and expert curation of a specific type of engineered part [21]. This type of resource needs to be extended to more types of genetic parts. Integrating work on documenting part variants using

a micropublication [57, 58] or wiki model [59] could be ways to recognize the contributions of curators and researchers to this kind of resource, hopefully including those with first-hand knowledge of the histories of important genetic parts. In the end, a combination of computational and community-based curation efforts will likely be the most effective path forward for improving plasmid annotation.

## Conclusions

As fully sequencing engineered plasmids becomes commonplace, researchers are encountering an overwhelming number of uncatalogued variants of canonical genetic parts and being forced to reckon with whether these differences are important or not. We developed a procedure for predicting variants that are likely to have arisen due to convergent evolution or engineering. We combined these predictions with genetic part variants that are found in plasmids from many labs, under the premise that both widespread and recurrent variants are more likely to affect the function of a genetic part and the reproducibility of research than random one-off changes. Genetic part variants in our final list of 217 predictions warrant further investigation and should be integrated into tools that annotate engineered DNA. This work is a promising step towards automating better plasmid annotation, but there is still a need for integrating this information with expert curation to create comprehensive databases of genetic parts.

## Materials and methods

### Identification of genetic part variants in engineered plasmids

We downloaded 51,359 complete plasmid sequences from Addgene, a non-profit plasmid repository based in Cambridge, Massachusetts, on August 9th, 2021. Plasmid sequences were annotated using pLannotate v1.2.0, which identifies matches to the Swissprot [60] (release 2021_03), Snapgene (2021-07-23), FPbase [21] (2020-09-02), and Rfam [61] (release 14.5) databases. We extracted all annotated features from every plasmid, keeping matches that pLannotate identified as covering $\geq$ 95% of the length of the feature in the database. Matches that were 100% identical at either the nucleotide or amino acid level to annotation database entries were removed. Protein-coding sequence features with 3′ or 5′ deletions were also removed. The remaining non-consensus features were considered genetic part variants and further analyzed.

### Grouping genetic part variants on related plasmids

The design similarity (DS) score is calculated based on a formula that is similar to that for the Inverse Document Frequency (IDF) of the most common segment shared by two plasmids, except extra terms are added when there are multiple segments shared by the two plasmids. For each genetic part variant found in plasmids from two or more depositing labs, we first performed a pairwise BLASTN search (BLAST 2.10.1+) [62] between all plasmids that contained that variant to identify shared plasmid segments. Each of these segments was then queried against the entire database using BLASTN to find the number of plasmids that contained the segment. The following BLASTN parameters were used in both cases: mismatch penalty −8, match reward 2, gap open penalty 4, gap extend penalty 6, and word size 28. These parameters were chosen to maximize the reporting of matches consisting of contiguous segments with few point mutations. A segment match was defined as having $\geq$98% identity, an E-value $\leq 10^{-5}$, and a length difference of at most 10 bp. The DS score was then calculated using the following equation:

$$\text{Design Similarity} = \log\left(\frac{p}{x_1} + \frac{\sum_{i=2}^{n} \frac{p}{x_i}}{n}\right)$$

*x* is a vector of length *n* containing the number of plasmids matching each segment query, sorted from the smallest to the largest value. *p* is the number of reference plasmids in the database. The rightmost term in the sum is an extra score heuristic that is applied when there is more than one matching segment.

We also cataloged all variants that were found in plasmids from $\geq$20 depositing labs, irrespective of DS. It was not computationally feasible to calculate pairwise DS scores for variants with >1,205 observations, but all 11 of these variants were catalogued because they were found on plasmids originating in $\geq$20 labs.

## Determining a threshold for plasmid relatedness

To determine a DS score threshold that indicates two examples of a genetic part variant on different plasmids likely shared an ancestor, we examined the distribution of DS scores for 100,000 random plasmid pairs. We picked only plasmid pairs that did not share a common depositing lab to increase the likelihood that we did not include pairs that did share a construction history in this set. We picked a DS cutoff for plasmid relatedness that gave a 5% false-positive rate on this dataset as the metric for calling two plasmids as related.

After calculating the pairwise DS scores for each group of plasmids that shared the same genetic part variant, we binarized the results based on the DS score cutoff threshold. The binary adjacency matrices were then analyzed as a network, and we counted the number of unlinked subgraphs within each plasmid network to estimate the number of times the variant had independently appeared.

## Supporting information

**S1 Table. Final list of 217 widespread and/or recurrent genetic part variants.**
(CSV)

## Acknowledgments

We thank members of the Barrick lab as well as Claus Wilke and his lab for helpful discussions and acknowledge the Texas Advanced Computing Center (TACC) at The University of Texas at Austin for providing high-performance computing resources.

## Author Contributions

**Conceptualization:** Matthew J. McGuffie, Jeffrey E. Barrick.

**Data curation:** Matthew J. McGuffie.

**Formal analysis:** Matthew J. McGuffie.

**Funding acquisition:** Jeffrey E. Barrick.

**Investigation:** Matthew J. McGuffie.

**Software:** Matthew J. McGuffie.

**Supervision:** Jeffrey E. Barrick.

**Visualization:** Matthew J. McGuffie.

**Writing – original draft:** Matthew J. McGuffie, Jeffrey E. Barrick.

**Writing – review & editing:** Matthew J. McGuffie, Jeffrey E. Barrick.

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
