## [Decision Letter · Decision Letter 0]

20 Mar 2024

PONE-D-24-05108Identifying widespread and recurrent variants of genetic parts to improve annotation of engineered DNA sequencesPLOS ONE

Dear Dr. Barrick,

Thank you for submitting your manuscript to PLOS ONE. After careful consideration, we feel that it has merit but does not fully meet PLOS ONE’s publication criteria as it currently stands. Therefore, we invite you to submit a revised version of the manuscript that addresses the points raised during the review process.

We look forward to receiving your revised manuscript.

Kind regards,

Bashir Sajo Mienda, PhD

Academic Editor

PLOS ONE

2. We note that you have referenced (Raciti D, Yook K, Harris TW, Schedl T, Sternberg PW. Micropublication: incentivizing community curation and placing unpublished data into the public domain. Database. 2018;2018: bay013. doi:10.1093/database/bay013) which has currently not yet been accepted for publication. Please remove this from your References and amend this to state in the body of your manuscript: (ie “Bewick et al. [Unpublished]”) as detailed online in our guide for authors

Reviewers' comments:

Reviewer's Responses to Questions

**Comments to the Author**

1. Is the manuscript technically sound, and do the data support the conclusions?

Reviewer #1: Yes

Reviewer #2: Yes

2. Has the statistical analysis been performed appropriately and rigorously? 

Reviewer #1: N/A

Reviewer #2: Yes

3. Have the authors made all data underlying the findings in their manuscript fully available?

Reviewer #1: Yes

Reviewer #2: Yes

4. Is the manuscript presented in an intelligible fashion and written in standard English?

Reviewer #1: Yes

Reviewer #2: Yes

5. Review Comments to the Author

Reviewer #1: This manuscript addresses a common problem in molecular biological engineering communities about how well we know our plasmids. Using pLannotate pipeline that Author's lab recently developed, Authors discovered that there are uncharacterised genetic variations in the plasmids deposited in public depository database. The manuscript describes a finding in DNA sequence evolution in laboratories, which has been ignored broadly by the communities. This finding may explain a degree of phenotype variations in the characterisation works among different laboratories. Overall, this manuscript is acceptable.

However, it would be nice to address the following comment:

(1) Could Authors cut the result session into several sub-sessions to address a specific argument point per sub-session? In the current version, there is only one session-the whole result, but five figures. It is quite difficult to read through the whole result session with a clear mind.

Reviewer #2: In this manuscript, the authors use a bioinformatics approach to analyse a large collection of cloning and expression vectors to investigate variation in the different ‘genetic parts’ (i.e. components of the vector). Their overall motivation is to identify different part variants, so as to enable such variants to be annotated in plasmid sequences, and to signpost future research to test whether this variation affects the properties of that part. The research is valuable and thoughtfully done, and my suggestions are relatively minor:

- Figure 1. Please check the legend, since it appears to be incorrect (e.g. referring to orange triangles). I was also a bit confused by the distinction between ‘total variants’ and ‘unique variants’ — I assume that the latter refers to variants represented only once in the dataset? Or does ‘total variants’ imply some higher-level classification e.g. promoter types (Plac, PrrnB1 etc), and ‘unique variants’ refer to variants within this perhaps varying by only single bp — but in this case why are there fewer unique variants than total variants? Some clarification would be beneficial, and perhaps a cartoon describing the process would be a good way of doing this.

- Line 255. Was it exactly 7,500,000 pairwise comparisons, and why was this number chosen?

- It would be interesting to construct and visualise phylogenetic trees for some of the key components that have shown potential evolution, but this may not be possible, and is certainly not necessary for the current manuscript.

6. PLOS authors have the option to publish the peer review history of their article (what does this mean?). If published, this will include your full peer review and any attached files.

Reviewer #1: No

Reviewer #2: No

---

## [Author Response · Author response to Decision Letter 0]

7 Apr 2024

Please see the uploaded document.

---

## [Decision Letter · Decision Letter 1]

8 May 2024

Identifying widespread and recurrent variants of genetic parts to improve annotation of engineered DNA sequences

PONE-D-24-05108R1

Dear Dr. BARRICK,

We’re pleased to inform you that your manuscript has been judged scientifically suitable for publication and will be formally accepted for publication once it meets all outstanding technical requirements.

Kind regards,

Bashir Sajo Mienda, PhD

Academic Editor

PLOS ONE

Additional Editor Comments (optional):

Reviewers' comments:

Reviewer's Responses to Questions

**Comments to the Author**

1. If the authors have adequately addressed your comments raised in a previous round of review and you feel that this manuscript is now acceptable for publication, you may indicate that here to bypass the “Comments to the Author” section, enter your conflict of interest statement in the “Confidential to Editor” section, and submit your "Accept" recommendation.

Reviewer #1: All comments have been addressed

2. Is the manuscript technically sound, and do the data support the conclusions?

Reviewer #1: Yes

3. Has the statistical analysis been performed appropriately and rigorously? 

Reviewer #1: N/A

4. Have the authors made all data underlying the findings in their manuscript fully available?

Reviewer #1: Yes

5. Is the manuscript presented in an intelligible fashion and written in standard English?

Reviewer #1: Yes

6. Review Comments to the Author

Reviewer #1: My comments have been addressed. I do not have further comments to provide. All the best and congratulations.

7. PLOS authors have the option to publish the peer review history of their article (what does this mean?). If published, this will include your full peer review and any attached files.

Reviewer #1: No

---

## [Editor Report · Acceptance letter]

14 May 2024

PONE-D-24-05108R1 

PLOS ONE

Dear Dr. Barrick, 

I'm pleased to inform you that your manuscript has been deemed suitable for publication in PLOS ONE. Congratulations! Your manuscript is now being handed over to our production team.

Kind regards, 

on behalf of

Dr. Bashir Sajo Mienda 

Academic Editor

PLOS ONE